# Additives Imparting Antimicrobial Properties to Acrylic Bone Cements

**DOI:** 10.3390/ma14227031

**Published:** 2021-11-19

**Authors:** Alina Robu, Aurora Antoniac, Elena Grosu, Eugeniu Vasile, Anca Daniela Raiciu, Florin Iordache, Vasile Iulian Antoniac, Julietta V. Rau, Viktoriya G. Yankova, Lia Mara Ditu, Vicentiu Saceleanu

**Affiliations:** 1Faculty of Material Science and Engineering, UniversityPolitehnica of Bucharest, 313 Splaiul Independentei Street, District 6, 060042 Bucharest, Romania; alinarobu2021@gmail.com (A.R.); antoniac.aurora@gmail.com (A.A.); elena_grosu@yahoo.com (E.G.); antoniac.iulian@gmail.com (V.I.A.); 2Faculty of Applied Chemistry and Material Science, University Politehnica of Bucharest, 313 Splaiul Independentei Street, District 6, 060042 Bucharest, Romania; eugeniuvasile@yahoo.com; 3Faculty of Pharmacy, Titu Maiorescu University, 22 Dambovnicului Street, District 4, 040441 Bucharest, Romania; daniela_raiciu@yahoo.com; 4S.C. Hofigal S.A., 2 Intrarea Serelor Street, District 4, 042124 Bucharest, Romania; 5Faculty of Veterinary Medicine, University of Agronomic Sciences and Veterinary Medicine Bucharest, 59 Bulevardul Marasti, 050097 Bucharest, Romania; floriniordache84@yahoo.com; 6Academy of Romanian Scientists, 54 Splaiul Independentei Street, District 5, 050094 Bucharest, Romania; 7Istituto di Struttura della Materia, Consiglio Nazionale delle Ricerche (ISM-CNR), Via del Fosso del Cavaliere 100, 00133 Rome, Italy; 8Institute of Pharmacy, Department of Analytical, Physical and Colloid Chemistry, I.M. Sechenov First Moscow State Medical University, 8 Trubetskaya Street, Build. 2, 119991 Moscow, Russia; yankova_v_g@staff.sechenov.ru; 9Faculty of Biology, University of Bucharest, Intrarea Portocalelor Street, no. 1-3, 060101 Bucharest, Romania; 10Faculty of Medicine, University Lucian Blaga Sibiu, 2A Lucian Blaga Street, 550169 Sibiu, Romania; vicentiu.saceleanu@ulbsibiu.ro

**Keywords:** PMMA bone cements, peppermint essential oil, silver nanoparticles, gentamicin, antimicrobial properties

## Abstract

PMMA bone cements are mainly used to fix implanted prostheses and are introduced as a fluid mixture, which hardens over time. The problem of infected prosthesis could be solved due to the development of some new antibacterial bone cements. In this paper, we show the results obtained to develop four different modified PMMA bone cements by using antimicrobial additives, such as gentamicin, peppermint oil incorporated in hydroxyapatite, and silver nanoparticles incorporated in a ceramic glass matrix (2 and 4%). The structure and morphology of the modified bone cements were investigated by SEM and EDS. We perform experimental measurements on wettability, hydration degree, and degradation degree after immersion in simulated body fluid. The cytotoxicity was evaluated by MTT assay using the human MG-63 cell line. Antimicrobial properties were checked against standard strains *Staphylococcus aureus*, *Pseudomonas aeruginosa*, and *Candida albicans*. The addition of antimicrobial agents did not significantly affect the hydration and degradation degree. In terms of biocompatibility assessed by the MTT test, all experimental PMMA bone cements are biocompatible. The performance of bone cements with peppermint essential oil and silver nanoparticles against these two pathogens suggests that these antibacterial additives look promising to be used in clinical practice against bacterial infection.

## 1. Introduction

Bone cements are used to fix artificial joints, such as hip or knee prostheses. These materials fill the space between the implant and the joint and are introduced as a fluid mixture, which hardens over time. Bone cements can also be defined as a mixture of substances, a family of materials consisting of a powder phase and a liquid phase which, after mixing and homogenization, forms a paste which has the ability to cure, and self-stabilizing once implanted in the body. This material has a special flexibility and modelling, which ensures the fixation of the material at the implant site and a good bone-material contact, even in complex defects from a geometric point of view [1].

Acrylic bone cements are polymeric materials, which are obtained by a polymerization reaction and produce a stable, non-absorbable material [2,3,4]. Poly(methyl methacrylate) (PMMA) based bone cements are two-component systems, comprising a solid phase (a polymeric powder) and a liquid phase (a liquid monomer). It also contains an inhibitor (hydroquinone) to prevent premature polymerization during storage and optionally a colouring agent, such as chlorophyll [5].

An implanted prosthesis is particularly sensitive to the development of bacteria on its surface because microorganisms can multiply as an adverse reaction of the body’s immune system on a foreign body or by primary contamination due to non-sterile materials or by hematogenous spread, where the infection comes from another infected area of the body and is transferred to the implant area. As the bacteria reproduce, they form a protective biofilm, which has a low sensitivity to antibiotics [6]. The release of antibiotics from the cement matrix takes place through surface diffusion, bulk diffusion, pores, and defects in the cement start. The mixing method, particle size, and uniformity of distribution significantly influence the antibiotic release kinetics [7].

Bacteria that can cause periprosthetic joint infections have a broad spectrum, most often caused by Coagulase negative staphylococci (in 30–43% of cases), followed by *Staphylococcus aureus* (12–23%) and Streptococci (9–10%). Other bacteria encountered in rare cases that can cause infections are represented by Enterococci, Gram-negative bacilli, anaerobic, polymicrobials, or others unknown microbes [8,9]. Currently, research in this field has gathered the best evidence for an adequate selection of antibiotics available especially for staphylococcal species, while, for other bacteria, such as streptococci, enterococci, or Gram-negative, the selection of antibiotics is more complex [10]. Different antibacterial agents were used by researchers for obtaining modified PMMA bone cements with antibacterial properties: antibiotics [11,12,13,14,15,16,17,18,19,20,21], silver nanoparticles [22,23,24,25], gold nanoparticles [26], hydroxyapatite [27,28,29,30,31], graphene [32,33], essential oil [34,35,36,37,38,39,40,41,42], or essential oils incorporated in different materials [43,44,45,46]. A schematic representation of cement preparation procedure is shown in Figure 1.

Gentamicin (GM) is the most commonly used antibiotic for treating bacterial infections that has a broad spectrum, thermal stability, and high solubility in water [11]. It is an aminoglycoside whose antibacterial activity is concentration dependent [12]. Other common antibiotics, such as tobramycin [13] and vancomycin [14], have also been used to treat periprosthetic joint infections. In order to increase the antimicrobial spectrum and, thus, the effectiveness of the treatment of periprosthetic joint infections, researchers tested the effect of acrylic bone cements with gentamicin + vancomycin + linezoid [15], gentamicin + teicoplanin [16], vancomycin + daptomycin [17], and vancomycin + cefazolin [18], which provide satisfactory results on their antibacterial activity.

However, the widespread use of antibiotics has led to the development of genetic and biochemical mechanisms that allow bacteria to survive in antibiotic environments. This is a cause for concern regarding the effectiveness of antibiotics commonly used in the composition of bone cements, especially the effectiveness of gentamicin. For example, in a study realized by Hellmark et al. [19] on joint prosthesis revision cases, a 79% resistance to gentamicin against a *Staphylococcus Epidermidis* was detected. Thornes et al. [20] confirmed these results with their research realized on rat models, which demonstrated an increased resistance of same bacteria to PMMA bone cement with gentamicin. Thus, the problem of low antibiotic efficacy in bone cement has created the need to investigate the potential to incorporate new antimicrobial agents into bone cements.

The availability of silver nanoparticles having a high surface to volume ratio and possessing which gives unique chemical and physical properties that greatly enhance the antimicrobial effects of silver. Silver ions inactivate enzymes vital to bacteria and disactivate the mechanism of bacterial DNA replication. Antimicrobial testing has also shown that cements with silver nanoparticles do not have antimicrobial activity against planktonic bacteria; however, they are able to quantitatively reduce the formation of bacterial biofilms on the surface of the cement. These results imply that cements containing silver nanoparticles have a high potential for use in arthroplasty interventions, in which it is necessary to prevent bacterial adhesion [22,23,24,25].

Other antimicrobial additives have also been studied to improve the antimicrobial activity of PMMA bone cements. For example, Russo et al. [26] used gold nanoparticles and demonstrated their antibacterial activity against Methicillin Resistant *Staphylococcus aureus* (MRSA) and *Pseudomonas aeruginosa* strains. Phakatkar et al. [27] introduced magnesium phosphate and hydroxyapatite into PMMA bone cement and obtained satisfying results on antibacterial properties and superior bioactivity related to cell viability due to the addition of hydroxyapatite. The antimicrobial effect of graphene embedded in acrylic resins used in the dental field is discussed in a review by Radhi et al. [32].

In the past couple of years, the use of plant-derived molecules in composition of some synthetic biomaterials for wound healing has started to gain widespread acceptance. Although the use of essential oils for medical applications is relatively new, it has raised interest being based on their long history of being used by humans for therapeutic purposes showing inhibition against bacteria (Gram negative and Gram positive) and fungi [34,35]. This interest in medication with straight source plants is due to the belief that natural components are safer and more dependable, compared with synthetic drugs that sometimes can have adverse effects [36]. Essential oils based on plant extracts, such as peppermint, cinnamon, lemon, and clove, have been applied to accelerate the wound healing process since ancient times due to their various therapeutic benefits, such as antibacterial activity, anti-inflammatory, and antioxidation potential [37]. However, the volatility, low stability, and high sensitivity to environmental factors have limited the usage of essential oils in wound healing applications, but, in the last decade, researchers have focused their attention on biomaterials containing essential oils to study their therapeutic properties [38].

A commonly considered essential oil is the one obtained from peppermint. Peppermint is mainly derived from the *Mentha piperita* leaf, and its antibacterial activity is due to its chemical composition of L-menthol, menthone, methyl acetate, and limonene [39,40]. Previous studies on the antimicrobial properties of organic or inorganic supports coated with peppermint essential oils reported similar results regarding their biological activity against pathogenic microorganisms [41,42]. Thus, the antibacterial activity of the hydroxyapatite coated with peppermint essential oil (HAp-P) were tested against different bacteria strains, such as *Staphylococcus aureus*, *Escherichia coli*, *Pseudomonas aeruginosa*, and *Candida albicans strains*, with the results highlighting that the antimicrobial activity of HAp-P increased significantly over that of hydroxyapatite alone [43]. Peppermint extract was also incorporated in the polyurethane (PU)-based nanofibers in order to be used in diabetic wound healing. The results showed that the release of extract from nanofibers continued for 144 h, with antibacterial efficiency of 99.9% against *Staphylococcus aureus* and *Escherichia coli* bacteria [44]. Kasiri and Fathi [45] studied the method for peppermint essential oil encapsulation with cellulose nanocrystals (CNC), loading capacity, and the *in vitro* release of the PEO, demonstrating that the main mechanism for release of peppermint oil is by diffusion transport.

The main objective of the paper was to develop novel modified PMMA bone cements with antimicrobial properties using hydroxyapatite impregnated with peppermint oil, silver nanoparticles incorporated in ceramic glass matrix, and gentamicin as antimicrobial agents. The novelty is consistent due to the antimicrobial agents, such as essential oil, incorporated in hydroxyapatite and silver nanoparticles incorporated in a ceramic glass used for bone cement. The SEM characterization, contact angle measurements, hydration, and degradation degree experiments were carried out. The *in vitro* tests were performed to demonstrate cements safety and efficacy. The biocompatibility was assessed by the MTT test using human MG-63 cell line. The antimicrobial tests of the developed new cement compositions against gram-positive *Staphylococcus aureus*, gram-negative *Pseudomonas aeruginosa*, and *Candida albicans* fungus microorganisms were performed. 

## 2. Materials and Methods

The experimental modified PMMA bone cements were obtained by modifying the solid component of commercial cement based on PMMA, with standard viscosity.

Different bone cements-type biomaterials were developed by combining different antimicrobial additives with the powder of the commercial PMMA bone cement, in order to determine the antimicrobial activity of the cement without negatively altering its mechanical performance. As antimicrobial agents, silver nanoparticles incorporated in a ceramic glass matrix (purchased from SANITIZED AG, Burgdorf, Switzerland) were used, respectively, hydroxyapatite (HAp, >95% purity, purchased from the Plasma Biotal Limited (Tideswell, UK)) impregnated with peppermint oil and gentamicin, being obtained 4 new bone cement compositions.

The chemical composition of commercial cement used as a reference and new antimicrobial bone cements are shown in Table 1 below.

**Scanning Electron Microscopy** (FEI Company, Eindhoven, Netherlands)coupled with Energy Dispersive Spectroscopy analysis on bone cements samples was performed using a QUANTA INSPECT F Scanning Electron Microscope (FEI Company, Eindhoven, Netherlands) equipped with Energy Dispersive X-ray Spectrometer Detector (EDAX) (FEI Company, Eindhoven, Netherlands) with a 132 eV resolution at MnKα. Analyses were performed in order to evaluate the morphology and antimicrobial additives distribution in the polymerized matrix.

The ***contact angle*** measurements were performed on the KRÜSS DSA30 Drop Shape Analysis System, in order to determine the wetting degree of solid and liquid interaction. The contact angle measurements were based on the sessile drop method, each measurement being repeated five times.

The ***hydration degree (Ha)*** was evaluated by immersing the experimental bone cement samples in distilled water at 37 °C for a period of 7, 14, and 21 days, respectively. At each time interval, the samples were removed from the distilled water, weighed and kept in a desiccator until they reached a constant mass. The degree of hydration (Ha) was determined using the following formula:(1)$$H_{a}\;\left(\%\right)=\left(\frac{m_{w}-m_{f}}{m_{0}}\right)\times 100,$$
where: *m_w_*—mass of the wet bone cement samples,*m_f_*—mass of the bone cement samples after drying, and*m*_0_—initial mass of the bone cement samples.

The ***degradation degree (DR)*** test involved the immersion of the bone cement samples in solution of simulated body fluid (SBF) at 37 °C for 1, 3, 5, 7, 14, 21, and 28 days, respectively. The SBF with an ionic composition similar to human blood plasma was prepared in the laboratory (SBF chemical composition: 8.035 g/L NaCl; 0.355 g/L NaHCO_3_; 0.225 g/L KCl; 0.231 g/L K_2_HPO_4_•3H_2_O; 0.311 g/L MgCl_2_•6H_2_O; 39 mL 1M HCl; 0.292 g/L CaCl_2_; 0.072 g/L Na_2_SO_4_; 6.118 g/L Tris-NH_2_C(CH_2_OH)_3_; substances were purchased from Sigma Aldrich, Taufkirchen, Germany). In addition, at each time interval, the samples were removed from the medium, weighed, and kept in a desiccator until they reached a constant mass. Degradation rates (DR) of the experimental samples were determined with the formula:(2)$$DR\;\left(\%\right)=\left(\frac{m_{0}-m_{f}}{m_{o}}\right)\times 100,$$
where: *m_f_*—mass of the bone cement samples after drying; and*m*_0_—initial mass of the bone cement samples.

***Antimicrobial tests.*** Microbial strains used in the experiments were represented by standard strains of *Staphylococcus aureus* ATCC 25923, *Pseudomonas aeruginosa* ATCC 27853, and *Candida albicans* ATCC 10231. To avoid the impact of contaminants on the experiment, the samples were previously sterilized by UV radiation (Benchmark Scientific, Sayreville, USA), for 30 min.

The qualitative screening was performed using an adapted spot diffusion method as a standardized method recommended to investigate the antimicrobial activity of different compounds, including antibiotics (according with Performance Standards for Antimicrobial Susceptibility Testing, Clinical and Laboratory Standard Institute 2021). Microbial suspensions of 1.5 × 108 CFU/mL (corresponding to 0.5 McFarland density), obtained from 24 h microbial cultures developed on corresponding agar media, were used in the experiments. Petri dishes with Muller Hinton agar (for bacterial strains) and Sabouraud agar (for yeast strain) were seeded with microbial inoculums. Subsequently, samples of bone cements with a diameter of 1 cm were placed on the surface of the medium and gently pressed to be in direct contact with the medium. After their free diffusion, the plates were incubated for 16–18 h at room temperature. The sensitivity of microbial strains was assessed by measuring the diameters of the inhibition zones around the bone cements samples and expressing them in “mm”.

***Bacterial adherence assay using viable cell count method.*** Quantitative assessment of the capacity of the selected strains to adhere on the surface of the tested samples was performed using the 24 multi-well plates. Overnight bacterial cultures of *Staphylococcus aureus* ATCC 25923, *Pseudomonas aeruginosa* ATCC 27853, and *Candida albicans* ATCC 10231 were diluted in fresh nutrient broth media at 1.5 × 105 CFU/mL final density. Two millilitres of the obtained suspension were seeded in 24 multi-well plates containing the treated materials, previously sterilized by UV irradiation. The plates were incubated at 37 °C, for 24, 48, and 72 h, the period during which bacterial cells are multiplied and, after reaching a threshold density, begin to adhere to the surface of the treated material and generate monospecific biofilm. For the adherence assay, after the incubation period, the materials were gently washed with sterile phosphate buffered saline (PBS, Sigma-Aldrich, St. Louis, MO, USA) in order to remove the non-adherent microbial cells and placed in 14 mL centrifuge tubes containing 1 mL of sterile PBS. The samples were vigorously mixed by vortexing for 1 min and sonicated for 10 s. Serial dilutions obtained from each sample were inoculated on LB agar plates in triplicates, and viable cell counts (VCCs) were assessed after incubation at 37 °C, for 24 h [34].

***Evaluation of biocompatibility properties of experimental bone cements samples.*** The human MG-63 cell line (ATCC CRL-1427, Manassas, VA, USA, Sigma-Aldrich) were used to evaluate the biocompatibility of 5 types of bone cement samples. Human MG-63 cell line is a stable Human osteosarcoma cell line and provides representative models to study the cytotoxicity of acrylic bone cement extracts to fabricate three-dimensional (3-D) niches and to study the cell viability, adhesion, and proliferation (product recommendations) [47]. The cells were cultured in Dulbecco′s Modified Eagle′s Medium (DMEM) medium (Sigma-Aldrich, MO, USA) supplemented with 10% fetal bovine serum, 1% penicillin, and 1% streptomycin antibiotics (Sigma-Aldrich, St. Louis, MO, USA). To maintain optimal culture conditions, medium was changed twice a week. The biocompatibility was assessed using MTT assay (Vybrant^®^ MTT Cell Proliferation Assay Kit, Thermo Fischer Scientific, Waltham, MA, USA). The viable cells reduce yellow tetrazolium salt MTT (3-(4,5 dimethylthiazole)-2,5-diphenyltetrazolium bromide) to a dark blue formazan via mitochondrial enzymes. Briefly, MG-63 cells were grown in 24-well plates, with a seeding density of 10.000 cells/well in the presence of bone cements for 72 h. Then, 15 mL of Solution I was added and incubated at 37 °C for 4 h. Solution II was added and pipettes vigorously to solubilise formazan crystals. After 1 h, the absorbance was read using spectrophotometer at 570 nm (TECAN Infinite M200, Männedorf, Switzerland).

***Fluorescent microscopy for tracing of living cells***. The biocompatibility between the bones cements samples and human MG-63 cell line was also evaluated based on fluorescent microscopy using RED CMTPX fluorophore (Thermo Fischer Scientific, MA, USA). The CMTPX tracker was added in cell culture in the presence of bone cements, and the viability and morphology of the cells was evaluated after 5 days. The CMTPX fluorophore was added in the culture medium at a final concentration of 5 μM and incubated for 30 min in order to allow the dye penetration into the cells. Next, the cells were washed with PBS and visualized by fluorescent microscopy. The photomicrographs were taken with Olympus CKX 41 digital camera driven by Cell Sense Entry software (Olympus, Tokyo, Japan).

***Alizarin Red assay.*** To quantify the osteogenic potential of bone cements samples, MG-63 cells were cultivated for 21 days in the presence of bone cement samples and stained with Alizarin Red dye. The cells were fixed with paraformaldehyde (4%, 2 h), washed three times with PBS, and then incubated with 40 nmol/L Alizarin Red S. This dye binds to calcium crystals in cells or matrix fibres, revealing a red colour. Unbound solution was washed out under gentle rocking. Bound dye was then released from the cells using 10% acetic acid incubation (1h), followed by neutralization with 10% ammonium hydroxide. The concentration of dye in solution was then quantified using absorbance spectroscopy at 405 nm wavelength (TECAN Infinite M200, Männedorf, Switzerland).

## 3. Results and Discussion 

### 3.1. SEM Analysis

In Figure 2, the morphologies of the obtained bone cements sample revealed by SEM microscopy images and corresponding EDS spectra are presented. The SEM analysis revealed that, after hardening, all the bone cements materials displayed the well-known microstructure typical for PMMA-based cements, in which four phases are highlighted: beads from the polymer powder; pores; the polymerized monomer; and the radiopacifying element, in this case, barium sulphate (BaSO_4_) [48,49,50,51]. SEM images show a good dispersion of the barium sulphate and antimicrobial additives into the PMMA-PBMA matrix (poly(methyl methacrylate)-poly(butyl methacrylate) matrix). A slight tendency of antimicrobial agents to form agglomerates can be observed, mostly in the bone cements samples containing 4% silver nanoparticles (sample AM2). The EDS analysis supports this observation and highlights the presence of the majority elements, C and O, from the two polymer phases; Ba and S from the composition of the radiopacifying element, respectively; and Ag, P, Ca, and Mg from HAp and silver nanoparticles incorporated in a ceramic glass matrix.

### 3.2. Contact Angle

The behaviour regarding the wettability of the surface is influenced by the type of antimicrobial additives used, as can be seen in Figure 3. For testing, 3 types of liquids were used, namely: water, di-iodomethane (DIM), and ethylene glycol (EG). The behaviour regarding the wettability of the surface is influenced by both the type of sample and the type of liquid used, as can be seen in Table 2. The most hydrophobic character can be observed in the case of the GM sample, which is more accentuated than in the case of the reference sample. The HUM, AM1, and AM2 samples have a more hydrophilic character, which is caused by the ceramic components (hydroxyapatite and ceramic glass) that have been added with the respective antimicrobial agents. It is known as the optimal contact angle value for cell adhesion 55° and for bone regeneration values between 35–80° [52,53]. All additives used kept the contact angle values within the desired limits so that they remain hydrophilic.

### 3.3. Hydration Degree 

Water absorption can influence the mechanical properties of bone cement because the fluid acts as a plasticizer inside the polymer matrix. In addition, water absorption can lead to the dissolution and release of water-soluble materials from the bone cement composition, such as antimicrobial agents or residual monomer. The absorbed water can lighten internal strains and allow the extraction of residual monomer and added additives. Moreover, water can penetrate more easily into a porous, heterogeneous structure with the interface area between the polymer and filler well-defined. The more heterogeneous areas in the structure of the material are the many spaces through which water can be accumulated. Figure 4 graphically represents the evolution of the hydration degree (Ha) evaluated in distilled water at a temperature of 37 °C over a period of 21 days, for all experimental samples obtained from bone cements based on PMMA.

The results show an increase in the hydration degree (Ha) of bone cements with the addition of antimicrobial agent and the amount of antimicrobial agent in AM1 and AM2 samples. Compared to the standard sample R, the values of the hydration degree (Ha) increased less in the case of the sample with gentamicin (GM sample). The largest amount of water is subsequently absorbed during the first 7 days after immersion; the values remain almost constant in the case of the reference R, GM, and HUM samples. A variation is observed in the case of samples containing silver nanoparticles in a dose-dependent manner. After exposure of the investigated bone cement samples to distilled water, no significant change in their structure was observed (Figure 5).

### 3.4. Degradation Degree

Figure 6 shows the variation of the degradation rate (DR) over a period of 28 days in simulated body fluid (SBF) at a temperature of 37 °C, for the investigated bone cements samples. After 28 days of exposure to SBF solution, the slight weight losses were observed for GM samples, while the reference sample (R) has a constant degradation rate (DR). An increase in the degradation rate was observed for samples containing peppermint essential oil as additive. Following the exposure to the SBF solution, some particles could have been removed from the matrix of bone cements or dissolved, which led to a reduction in weight. In the case of samples containing silver nanoparticles, up to 28 days, no degradation process is observed, but only a continuous absorption process.

Exposure of the investigated bone cement samples to the SBF solution did not significantly affect their structure. EDS analysis showed the presence of C, O, S, and Ba (Figure 7).

The results recorded for the hydration degree (Ha) and the degradation rate (DR) are consistent with the results obtained by evaluation of the wettability of surfaces by determining the contact angle.

### 3.5. Antimicrobial Tests Results

Placing the samples in contact with the culture medium inoculated with the cell suspensions allowed the qualitative evaluation of the antimicrobial effect. The experimental tests showed that bone cement sample loaded with gentamicin (GM) expressed a clear inhibition zone only towards bacterial strains, being used as positive control for bone cement samples loaded with other different antimicrobial agents. In the same time, the HUM sample (loaded with 5% peppermint essential oils active components) and AM2 sample (loaded with 4% antimicrobial additive) showed their inhibitory effect, generating clear inhibition zones with diameters compared to those of the GM control sample: 25 mm for *Staphylococcus aureus* ATCC 25923 and 20 mm for *Pseudomonas aeruginosa* ATCC 27853 (Figure 8). The lowest sensitivity was expressed by the *Candida albicans* ATCC 10231 fungus strains; in this case, all the tested samples, including GM control sample, do not show any inhibition zone (Figure 9).

Quantitative assessment of the capacity of the selected strains to adhere and generate biofilm on the surface of the tested samples was performed using viable cell count method. The CFU/mL values were determined for the biofilms developed in 24, 48, and 72 h, on the inert substrate represented by the bone cement samples. Analysing the data presented in Figure 10, it can be observed that the MIC values corresponding to the AM2 and HUM samples decreased significantly, by more than 4 logarithmic units, compared to the R control sample, for all selected bacterial strains and at all 3 incubation times, but did not reach the MIC values of GM control samples. These results show that, although the bacterial cells poorly adhered to the surface of the bone cement after 24 incubations, they could not develop a mature biofilm, and the decrease in CFU/mL values after 48 and 72 h of incubation was not significant compared to 24 h. Instead, the capacity of the fungus strain *Candida albicans* to adhere and generate biofilm was significantly inhibited, registering the lowest values of CFU/mL, starting with 24 h, being the most sensitive tested strain.

### 3.6. MTT Assay Results

The tested bone cement samples stimulated cellular metabolism, with a significant increase in proliferation compared to the control (CTRL). The MTT assay revealed that all bone cements tested had no cytotoxic effect, with the absorbances values being higher compared to the control sample. Moreover, GM caused the strongest increase in cell proliferation (236%), followed by the AM2 (167%), R (157%), HUM (151%), and AM1 (139%) at 24 h (Figure 11). After 48 h in the presence of bone cements, MG-63 cells remain viable, with cell proliferation being more uniform between the samples than that recorded at 24h. At 72 h, the highest proliferation rate was observed in HUM (160%), followed by AM2 (155%), R (29%), GM (129%), and AM1 (107%).

After 5 days, in the presence of bone cements, the MG-63 cells showed normal morphology with a fibroblast-like characteristic appearance. Fluorescence images showed that MG-63 cells were viable, and no dead cells nor cell fragments were observed. Moreover, the cells formed phylopodia to move and establish contacts with neighboring cells, suggesting that MG-63 cells exhibited an active phenotype (Figure 12).

The osteogenic potential of bone cements onMG-63 cells was quantified using Alizarin Red assay. This test is used to stain, or locate, calcium deposits in cells and tissues, when Alizarin Red binds to the calcium to form a pigment that is orange to red in colour.

After 21 days, in the presence of bone cements, the MG-63 cells have increased their osteogenic potential. This was demonstrated by an increase in calcium deposits in all samples compared with the control. The quantification of calcium deposits ranged between 0.034 in control and 0.086 in AM2. The other cements have the following values: 0.072 for R, 0.073 for AM1, 0.082 for GM, and 0.074 HUM, respectively (Figure 13).

## 4. Conclusions

The addition of antimicrobial agents did not induce structural changes in the developed novel PMMA bone cement samples. All experimental specimens possessed a typical structure and morphology for PMMA bone cements, with a slight tendency to form agglomerates, mostly in the PMMA bone cements samples containing 4% silver nanoparticles. Wettability measured by contact angle decreased by adding the antimicrobial agents, except in the case of the PMMA bone cements modified with gentamicin. All additives used kept the contact angle values within the desired limits and showed good cell adhesion. Exposure of the experimental PMMA bone cement samples to the water and SBF solution did not significantly affect their structure. The addition of antimicrobial agents did not significantly affect the hydration and degradation degree of the PMMA bone cements.

The antimicrobial properties have been demonstrated for the GM, HUM, and AM2 samples, which generated clear inhibition zones towards *Staphylococcus aureus* ATCC 25923 and *Pseudomonas aeruginosa* ATCC 27853.It was observed that not only the type of antimicrobial agent is important but also the amount used. *Staphylococcus aureus* and *Pseudomonas aeruginosa* are known to be clinically relevant pathogens associated with bone infection and intrinsic drug resistance, respectively. In terms of biocompatibility assessed by the MTT test using human MG-63 cells, all cements are biocompatible without any cytotoxicity effect. The performance of bone cements with peppermint essential oil and silver nanoparticles against these two pathogens suggests that these antibacterial additives look promising to be used in clinical practice against bacterial infection.

## Figures and Tables

**Figure 1 materials-14-07031-f001:**
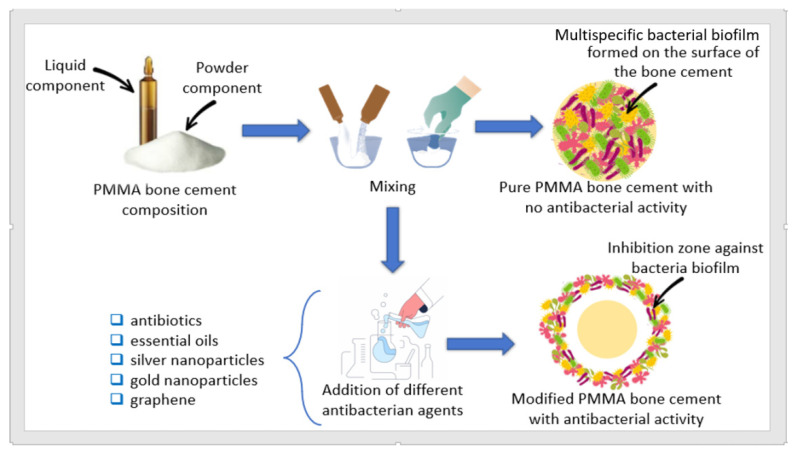
Schematic representation of the modified PMMA bone cements preparation procedure with antibacterial properties.

**Figure 2 materials-14-07031-f002:**
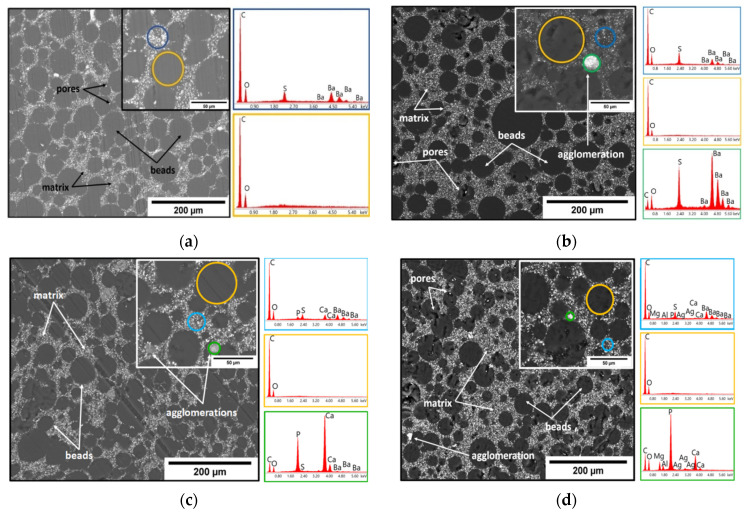
Representative SEM images and corresponding EDS spectra obtained from the highlighted areas of the experimental samples: (**a**) R sample; (**b**) GM sample; (**c**) HUM sample; (**d**) AM1 sample; (**e**) AM2 sample.

**Figure 3 materials-14-07031-f003:**
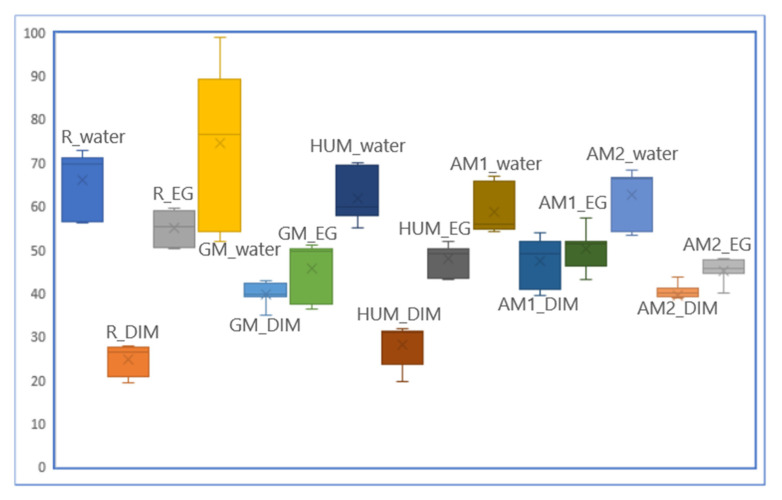
Surface wettability of the investigated bone cement samples determined by the measurements of the contact angle with distilled water, DIM, and EG.

**Figure 4 materials-14-07031-f004:**
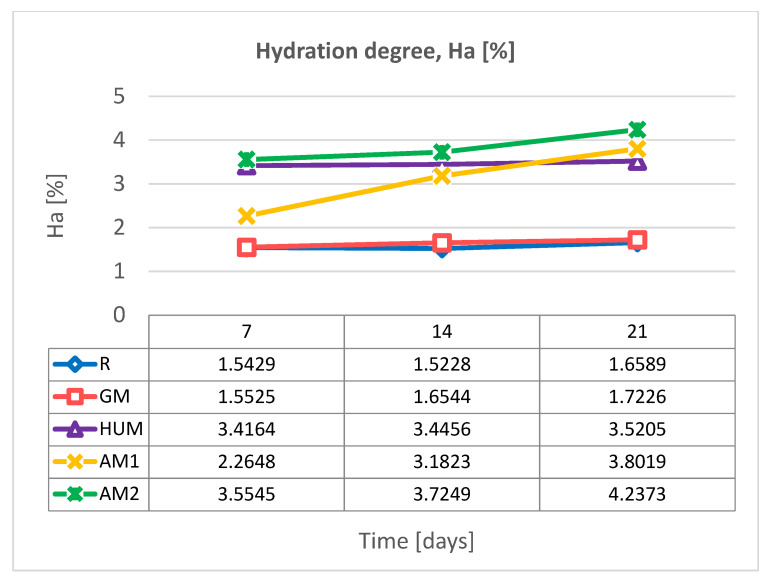
Variation of hydration degree over a period of 21 days in distilled water, at 37 °C, for experimental bone cements samples.

**Figure 5 materials-14-07031-f005:**
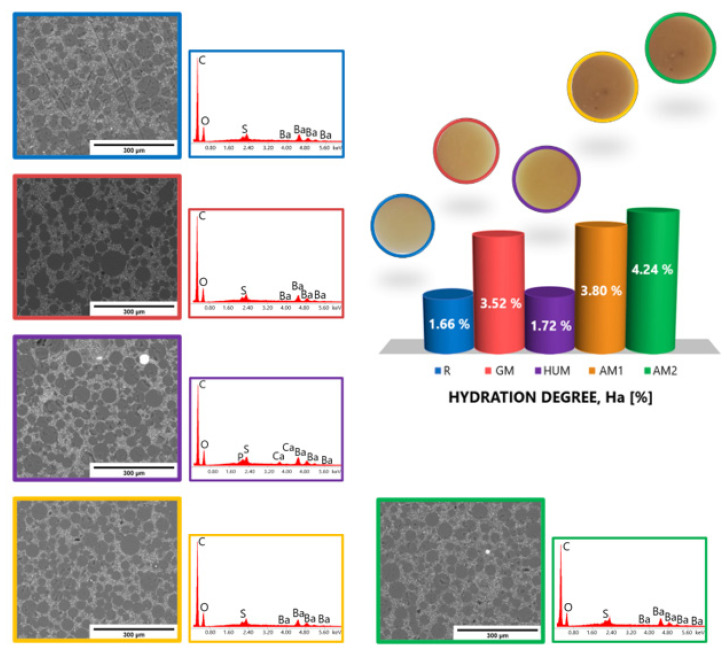
Morphological aspects revealed by SEM and elemental surface analysis by EDS of the samples after 21 days in distilled water.

**Figure 6 materials-14-07031-f006:**
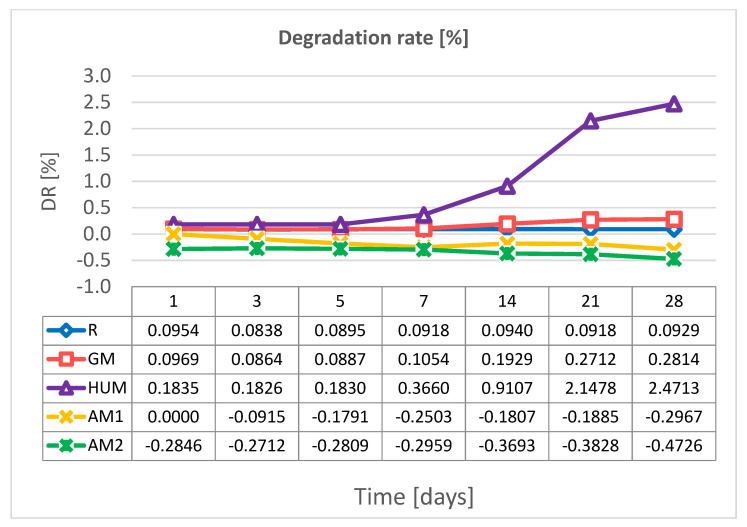
Variation of degradation rate over a period of 28 days in SBF, at 37 °C, for experimental PMMA bone cements.

**Figure 7 materials-14-07031-f007:**
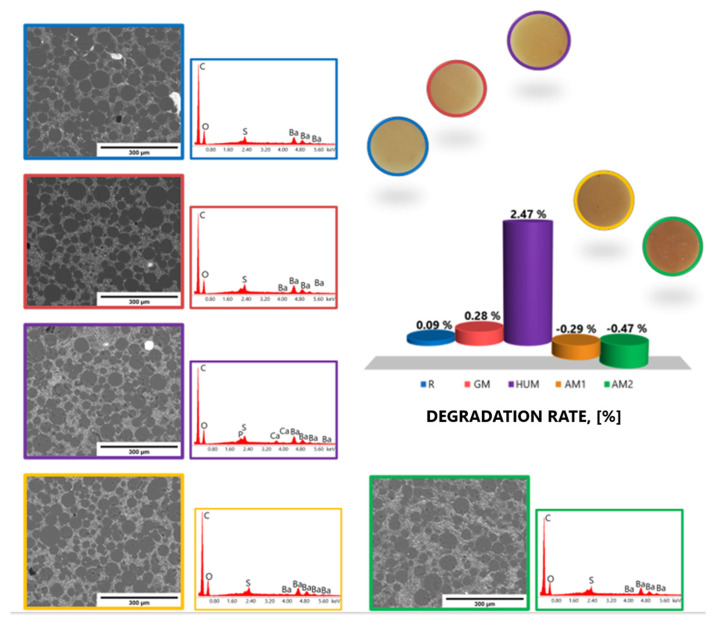
Morphological aspects revealed by SEM and elemental surface analysis by EDS of the experimental PMMA bone cements after 28 days in SBF.

**Figure 8 materials-14-07031-f008:**
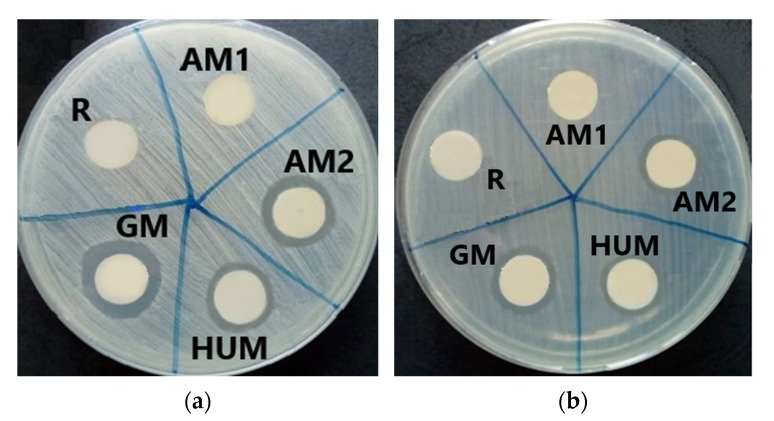
Antibacterial property evaluation for the experimental PMMA bone cements: (**a**) growth inhibition zone for Gram-positive strain *Staphylococcus aureus* ATCC 2592; (**b**) growth inhibition zone for Gram-negative strain *Pseudomonas aeruginosa* ATCC 27853, after 72 h of incubation.

**Figure 9 materials-14-07031-f009:**
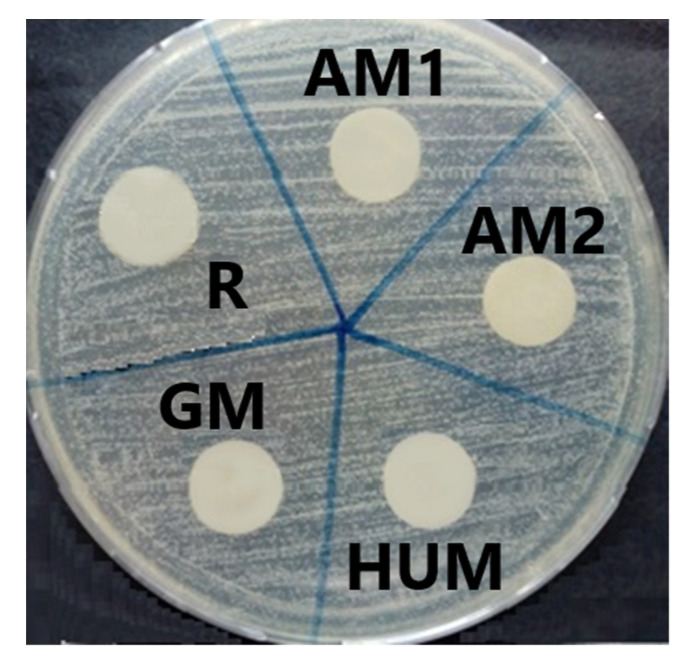
Contact of experimental PMMA bone cements towards *Candida albicans* ATCC 10231.

**Figure 10 materials-14-07031-f010:**
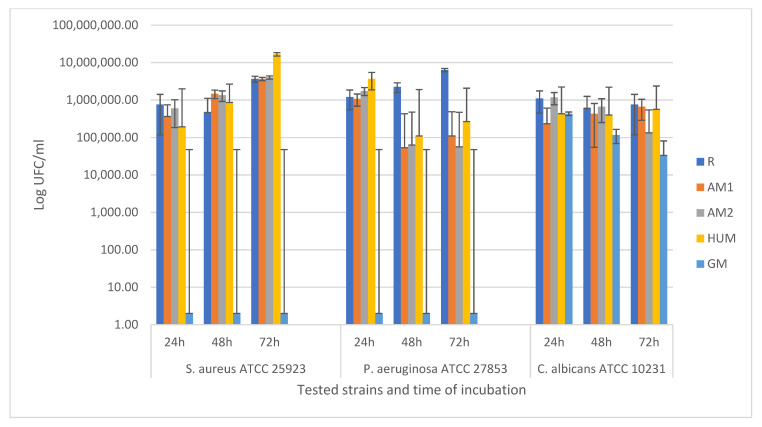
Graphical representation of CFU/mL values evaluating the ability of the tested strains to adhere and to develop monospecific biofilm in 24, 48, and 72h, on the surface of the experimental PMMA bone cements.

**Figure 11 materials-14-07031-f011:**
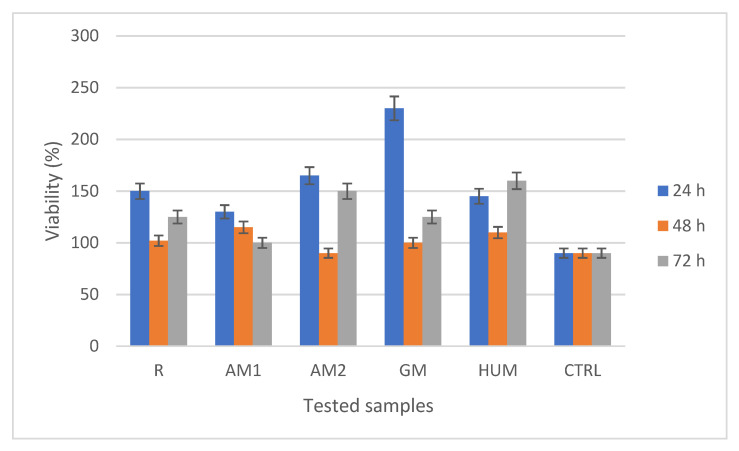
MTT assay showing the viability of MG-63 cells in the presence of the experimental PMMA bone cements after 24, 48, and 72 h.

**Figure 12 materials-14-07031-f012:**
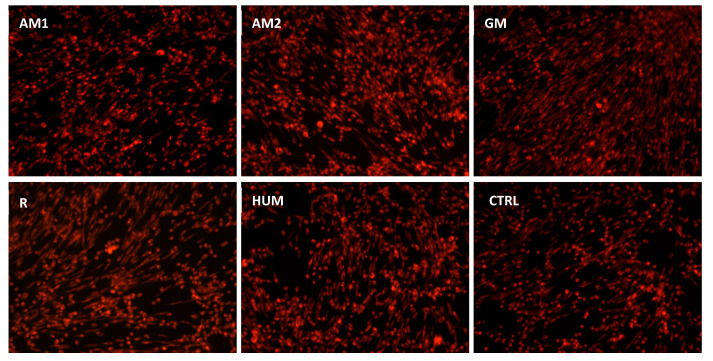
Fluorescence images of MG-63 cells coloured with CMTPX fluorophore in the presence of experimental PMMA bone cements.

**Figure 13 materials-14-07031-f013:**
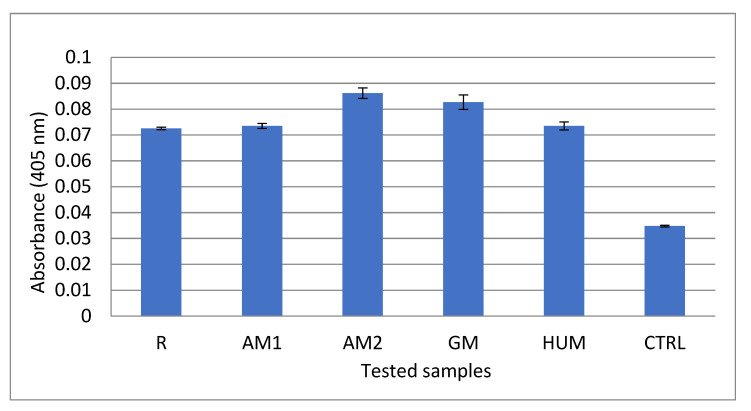
Alizarin Red assay showing the osteogenic potential of experimental PMMA bone cements on MG-63 cells.

**Table 1 materials-14-07031-t001:** Composition of the standard and new antimicrobial bone cements.

Samples	Composition		Antimicrobial Additive
	Powder	Liquid	
**R**	PMMA, BPO, BaSO_4_	MMA, BMA, DmpT, HQ	None
**GM**	PMMA, BPO, BaSO_4_	MMA, BMA, DmpT, HQ	5% gentamicin
**HUM**	PMMA, BPO, BaSO_4_, HAp	MMA, BMA, DmpT, HQ	5% peppermint essential oil incorporated in hydroxyapatite
**AM1**	PMMA, BPO, BaSO_4_	MMA, BMA, DmpT, HQ	2% silver nanoparticles incorporated in a ceramic glass
**AM2**	PMMA, BPO, BaSO_4_	MMA, BMA, DmpT, HQ	4% silver nanoparticles incorporated in a ceramic glass

Methyl methacrylate—MMA; Butyl methacrylate—BMA; *N*,*N*-Dimethyl-*p*-toluidine—DmpT; Hydroquinone—HQ; Poly(methyl-methacrylate)—PMMA, Benzoyl peroxide—BPO, Barium sulphate—BaSO_4_; Hydroxyapatite—HAp.

**Table 2 materials-14-07031-t002:** Contact angle values for experimental bone cement samples.

	R	GM	HUM	AM1	AM2
Contact angle, Ɵ [°], water	71	85	58	55	67
Contact angle, Ɵ [°], DIM	27	40	31	51	41
Contact angle, Ɵ [°], EG	57	50	50	53	46

## Data Availability

Not applicable.

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
