# Peer review of "Additives Imparting Antimicrobial Properties to Acrylic Bone Cements"

_materials, 2021, doi:10.3390/ma14227031_

Round 1

Reviewer 1 Report

This article needs some modifications to be suitable for this journal. I suggest minor revision for this paper. The main comments are:

  1. The paper is technically interesting; however, the novelty of this paper should be further justified and to establish the contributions to the new body of knowledge.
  2. Abstract section should be improved considering the following structure: introduction, problem statement, methodology, results, and conclusion.
  3. In Introduction section, the authors should improve the research background, the review of significant works in the specific study area, the knowledge gap, the problem statement, and the novelty of the research.
  4. Introduction is not organized well. The reviewer cannot readily see the significance of this study compared to the previous works. The literature review should be extended to more recently published works available in the literature. Some of the papers for authors which used advanced scanning electron microscopic technology should be refereed:

(a) Influence of super absorbent polymer on mechanical, rheological, durability, and microstructural properties of self‐compacting concrete using non‐biodegradable granite pulver, Structural Concrete 22, E1093-E1116

(b) Effect of super absorbent polymer on microstructural and mechanical properties of concrete blends using granite pulver, Structural Concrete 22, E898-E915

  1. The presentation of the results and conclusions could be improved.

Author Response

            Dear reviewer,

             The Authors of the manuscript entitled “Additives imparting antimicrobial properties to acrylic bone cements” submitted to MATERIALS thank the reviewer for reviewing our manuscript. We are deeply grateful for the observations and comments which we addressed and feel that greatly increased the quality of our manuscript. Please find below the answers to all comments and suggestions.

  1. The paper is technically interesting; however, the novelty of this paper should be further justified and to establish the contributions to the new body of knowledge.

Answer:  Thank you for your suggestion. The novelty of the work was mentioned clearly as follows: "The novelty is consistent due to the antimicrobial agents like essential oil incorporated in hydroxyapatite and silver nanoparticles incorporated in a ceramic glass used for bone cement." 

  1. Abstract section should be improved considering the following structure: introduction, problem statement, methodology, results, and conclusion.

Answer: Thank you for your suggestion. The abstract was organized according to the suggestions.

  1. In Introduction section, the authors should improve the research background, the review of significant works in the specific study area, the knowledge gap, the problem statement, and the novelty of the research.

Answer:  Thank you for your suggestion. The Introduction section was organized according to the suggestions.

  1. Introduction is not organized well. The reviewer cannot readily see the significance of this study compared to the previous works. The literature review should be extended to more recently published works available in the literature. Some of the papers for authors which used advanced scanning electron microscopic technology should be refereed.

Answer:  We perform the modifications in order to mention clearly the significance of this study compared to the previous work. "The novelty is consistent due to the antimicrobial agents like essential oil incorporated in hydroxyapatite and silver nanoparticles incorporated in a ceramic glass used for bone cement."  New references have been introduced in the article accordingly, respectively some of the papers for authors which used advanced scanning electron microscopic technology. The reviewer must be happy with these modifications.

  1. The presentation of the results and conclusions could be improved.

Answer:  The presentation of the results and conclusions was improved.

Reviewer 2 Report

Your manuscript entitled "Additives imparting antimicrobial properties to acrylic bone cements" is interesting to be published in Materials. It shows the characterizations of PMMA bone cements for fixing implanted prostheses. My comments are only minor prior the publication in Materials:

  1. Tables 1 and 2 look confusing. I don't think that you need to separate those tables.  Composition is also almost the same for whole samples while the only difference is the additional HaP for HUM sample (you did not mention, what is HaP. I think that the composition can be written in the text and Antimicrobial additive can be added in the table (for R, you just put "None"). The arrangement of table 2 also looks clumsy.
  2. Not consistent about the abbreviations. For instance, degree of hydrogenation, you first put Ha, and later you put without abbreviation. Degradation degree should be abbreviated but you did not do it. 
  3. Figure 2 also looks ugly and the figure was separated between two pages.   

Author Response

Dear reviewer,

 The Authors of the manuscript entitled “Additives imparting antimicrobial properties to acrylic bone cements” submitted to MATERIALS thank the reviewer for reviewing our manuscript. We are deeply grateful for the observations and comments which we addressed and feel that greatly increased the quality of our manuscript. Please find below the answers to all comments and suggestions.

  1. Tables 1 and 2 look confusing. I don't think that you need to separate those tables.  Composition is also almost the same for whole samples while the only difference is the additional HaP for HUM sample (you did not mention, what is HaP). I think that the composition can be written in the text and Antimicrobial additive can be added in the table (for R, you just put "None"). The arrangement of table 2 also looks clumsy.

Answer:  Thank you for your suggestion. Tables 1 and 2 were organized into a single table. HAp is the abbreviation for hydroxyapatite and this was mentioned now in the text. The new organization of the table has been made.

  1. Not consistent about the abbreviations. For instance, degree of hydrogenation, you first put Ha, and later you put without abbreviation. Degradation degree should be abbreviated but you did not do it. 

Answer: Abbreviations have been introduced in the text according to the observation made by the reviewer.

  1. Figure 2 also looks ugly and the figure was separated between two pages.   Answer:  The figure 2 will appear in one page.

Reviewer 3 Report

In this article authors Robu et al. describe Additives imparting antimicrobial properties to acrylic bone cements.

In my opinion, some issues should be further address and I hope following comments could be helpful for improving their paper:

  1. Introduction: Authors need to improve introduction part. Its too lengthy, try to make it concise and to the point.
  2. Why The qualitative screening was performed using an adapted spot diffusion method?
  3. why The human MG-63 cell line were used to evaluate the biocompatibility?
  4. How water absorption can lead to the dissolution and release of water-soluble materials from the bone cement?
  5. Following the exposure to the SBF solution, some particles could have been removed from the matrix of bone cements or dissolved, How dissolved?
  6. Discussion is written very poorly, Please revised it again.
  7. There is too many grammatical and typos mistakes, please revised your manuscript

Author Response

Dear reviewer,

The Authors of the manuscript entitled “Additives imparting antimicrobial properties to acrylic bone cements” submitted to MATERIALS thank the reviewer for reviewing our manuscript. We are deeply grateful for the observations and comments which we addressed and feel that greatly increased the quality of our manuscript. Please find below the answers to all comments and suggestions.

  1. Introduction: Authors need to improve introduction part. Its too lengthy, try to make itconcise and to the point.

Answer: Authors have improved the introduction part.

  1. Why the qualitative screening was performed using an adapted spot diffusion method?

Answer: The diffusion method is a standardized method recommended to investigate the antimicrobial activity of different compounds, including antibiotics (according with CLSI: Clinical and Laboratory Standards Institute).

  1. Why the human MG-63 cell line were used to evaluate the biocompatibility?

Answer: Human MG-63 cell line is a stable Human osteosarcoma cell line and provides representative models to study the cytotoxicity of acrylic bone cement extracts to fabricate three-dimensional (3-D) niches and to study the cell viability, adhesion and proliferation (product recommendations)  (https://www.sigmaaldrich.com/RO/en/product/sigma/cb_86051601).

  1. How water absorption can lead to the dissolution and release of water-solublematerials from the bone cement?

Answer: In the case of bone cements, the absorbed water can lighten internal strains and allow the extraction of residual monomer and/or added additives. Moreover, water can penetrate more easily into a porous, heterogeneous structure with the interface area between the polymer and filler well-defined. The more heterogeneous areas in the structure of the material are, the more spaces are through which water can be accumulated. Thus, water penetrates through the pores of the material and produces the extraction and dissolution of some water-soluble components from bone cements.

  1. Following the exposure to the SBF solution, some particles could have been removed from the matrix of bone cements or dissolved, How dissolved?

Answer: Dissolution appears in the case of the bone cement sample with gentamicin, when the antibiotic molecules come into contact with water (solution of SBF) they dissolved fast.

  1. Discussion is written very poorly, Please revised it again.

Answer: Discussion part was revised.

  1. There is too many grammatical and typos mistakes, please revised your manuscript

Answer: Grammar was checked and looks fine. Anyway, these aspects will be solved with editors (“proof correction” step).
